

# A new molecular diagnostic tool for surveying and monitoring *Triops cancriformis* populations

Graham S. Sellers[1], Larry R. Griffin[2], Bernd Hänfling[1] and Africa Gómez[1]

[1] School of Environmental Sciences, University of Hull, Hull, United Kingdom
[2] Conservation Programmes Directorate, Wildfowl & Wetlands Trust, Slimbridge, United Kingdom

## ABSTRACT

The tadpole shrimp, *Triops cancriformis*, is a freshwater crustacean listed as endangered in the UK and Europe living in ephemeral pools. Populations are threatened by habitat destruction due to land development for agriculture and increased urbanisation. Despite this, there is a lack of efficient methods for discovering and monitoring populations. Established macroinvertebrate monitoring methods, such as net sampling, are unsuitable given the organism's life history, that include long lived diapausing eggs, benthic habits and ephemerally active populations. Conventional hatching methods, such as sediment incubation, are both time consuming and potentially confounded by bet-hedging hatching strategies of diapausing eggs. Here we develop a new molecular diagnostic method to detect viable egg banks of *T. cancriformis*, and compare its performance to two conventional monitoring methods involving diapausing egg hatching. We apply this method to a collection of pond sediments from the Wildfowl & Wetlands Trust Caerlaverock National Nature Reserve, which holds one of the two remaining British populations of *T. cancriformis*. DNA barcoding of isolated eggs, using newly designed species-specific primers for a large region of mtDNA, was used to estimate egg viability. These estimates were compared to those obtained by the conventional methods of sediment and isolation hatching. Our method outperformed the conventional methods, revealing six ponds holding viable *T. cancriformis* diapausing egg banks in Caerlaverock. Additionally, designed species-specific primers for a short region of mtDNA identified degraded, inviable eggs and were used to ascertain the levels of recent mortality within an egg bank. Together with efficient sugar flotation techniques to extract eggs from sediment samples, our molecular method proved to be a faster and more powerful alternative for assessing the viability and condition of *T. cancriformis* diapausing egg banks.

Corresponding author
Graham S. Sellers,
graham.s.sellers@gmail.com

## INTRODUCTION

The tadpole shrimp, *Triops cancriformis* (Bosc, 1801), is a large freshwater branchiopod of the order Notostraca native to Europe (*Hughes, 1997*; *Zierold, Hänfling & Gómez, 2007*). As with passively dispersed ephemeral pool specialists, such as other branchiopods and rotifers,

*T. cancriformis* has adaptations to persist over unpredictable drought periods. These organisms produce diapausing eggs resistant to environmental extremes that can remain dormant for decades, accumulating in the pool sediment to form 'egg banks' of future generations (*Brendonck & De Meester, 2003*). In this stage the eggs can be passively dispersed by animal vectors (*Thiéry, 1997*; *Green & Figuerola, 2005*; *Vanschoenwinkel et al., 2011*; *Muñoz et al., 2013*), colonising potential new habitats over great distances. In addition, a bet-hedging hatching strategy is an adaptive feature in the life history of aquatic invertebrates from ephemeral ponds (*Simovich & Hathaway, 1997*; *Allen, 2010*), including *Triops* (*Takahashi, 1976*). Not all eggs hatch in a given hydroperiod, some remain dormant until future hydroperiods so spreading reproductive risk over time (*Seger & Brockmann, 1987*). Finally, *T. cancriformis* has a rapid life cycle. Reproductive age is reached in as little as 12 days and egg laying individuals deposit numerous egg clutches for the remainder of their life span (*Feber et al., 2011*). Across the European distribution of *T. cancriformis*, populations exhibit differing sexual systems; southern populations contain similar proportions of males and females whereas those further north are mostly selfing hermaphrodites (*Zierold et al., 2009*).

Throughout Europe and the United Kingdom ephemeral pools have been lost to, and are increasingly at risk from, land development for agriculture and urbanisation (*Serrano & Serrano, 1996*; *Céréghino et al., 2008*; *Williams et al., 2010*). As such *T. cancriformis* is classified as endangered in many European countries (*Eder & Hödl, 2002*) and in the UK it is protected under Schedule 5 of the Wildlife and Countryside Act 1981 with a Biodiversity Action Plan (BAP) (*Feber et al., 2011*). The New Forest (Hampshire, Southern England) and the Wildfowl & Wetlands Trust Caerlaverock National Nature Reserve (Dumfriesshire, South West Scotland), rediscovered in 2004, are the two locations of remaining populations of *T. cancriformis* known in the British Isles. Both are remnants of a historically wider distribution recorded in the south and south west of England (*Fox, 1949*) and south west Scotland (*Balfour-Browne, 1909*; *Balfour-Browne, 1948*). These two UK populations are toward the northernmost extent of the species range and are comprised of hermaphroditic individuals (*Zierold et al., 2009*). Given the ephemerality and passive dispersal of *T. cancriformis*, it is likely that undiscovered *T. cancriformis* egg banks and populations exist across the British Isles (as suggested by *Adams et al., 2014*).

Surveying methods such as water column netting and kick-sampling are conventionally employed to identify and assess aquatic macroinvertebrate communities within a water body (*Williams et al., 2004*; *Stark, 1993*). Many variations of these methods have been used worldwide to study large branchiopods (*Martin, Christopher Rogers & Olesen, 2016*), including *Triops* (*Sassaman, Simovich & Fugate, 1997*; *Zierold, Hänfling & Gómez, 2007*). However they rely on finding adult individuals within a water body. Differences in abiotic factors and a pools hydroregime can result in long periods with no records of *T. cancriformis* even within a known population site using such standard methods (*Feber et al., 2011*). Alternative sampling methods, more suited for the ephemeral nature of *Triops* life history, target the diapausing eggs. As the viability of a *Triops* egg cannot be visually discerned, unlike with rotifers (*García-Roger, Carmona & Serra, 2005*), viability estimates rely on successful hatching of diapausing eggs. Rehydration and incubation of sediment containing diapausing *Triops* eggs has been used for the study of hatchlings in the laboratory (*Sassaman,*

*Simovich & Fugate, 1997*; *Obregón-Barboza, Maeda-Martínez & Murugan, 2001*; *Schön-brunner & Eder, 2006*; *Harper & Reiber, 2006*; *Zierold, Hänfling & Gómez, 2007*). Additionally, collected sediment can also be progressively sieved through finer meshes to isolate, identify and hatch the eggs of *Triops* and other species it contains (*Kuller & Gasith, 1996*). A further method for the isolation of eggs is that of sucrose flotation (*Gómez & Carvalho, 2000*). This is a very efficient method which substantially reduces the time needed to find eggs in sediment. Of these alternative sampling methods only the incubation of sediment has been used to discover new *T. cancriformis* populations within Britain. *Adams et al. (2014)* surveyed 86 pools consisting of both extant and historic *Triops* population locations on the Solway Firth, UK, including the WWT Caerlaverock National Nature Reserve. Despite the large effort involved, the study only produced three hatched *T. cancriformis* nauplii from two of the sampled sites, over a period in excess of 70 days. The study however did discover a new population of *T. cancriformis* on the Solway Firth. The current methods used are all confounded by the non-uniform hatching of *Triops* at the beginning of a hydroperiod. Hatching of *Triops* eggs is dependent upon simulating favourable hatching conditions in the laboratory (*Kuller & Gasith, 1996*; *Eder, Hödl & Gottwald, 1997*; *Schönbrunner & Eder, 2006*; *Kashiyama et al., 2010*) and some, if not all, of the eggs present could remain dormant as a bet-hedging strategy (*Takahashi, 1976*).

A molecular approach can be applied to the discovery and identification of *T. cancriformis* populations in the UK, removing the associated deficiencies of conventional surveying methods. DNA barcoding using 'universal' primers and sequencing has been used extensively for species identification (*Hebert et al., 2003*). Environmental DNA (eDNA) has been employed to monitor endangered freshwater biodiversity across Europe, including another notostracan species: *Lepidurus apus* (*Thomsen et al., 2012*). Given that DNA degrades rapidly after an organism's death (*Hofreiter et al., 2001*), amplification of a large DNA fragment could potentially be used to assess egg viability in aquatic invertebrates. A species-specific amplification technique applied to isolated diapausing *T. cancriformis* eggs, amplifying a suitably large region of mtDNA, could both determine egg viability and species identity. Such an approach would remove the uncertainty of bet-hedging giving more reliable estimates of *T. cancriformis* egg bank viability. Conversely, species-specific primers designed for much shorter fragments, associated with degenerated mtDNA, could be used simultaneously to identify degraded non viable eggs. Although small fragments of DNA can persist *post mortem* for long periods of time this preservation requires rapid and prolonged desiccation or very low temperatures (*Lindahl, 1993*; *Hofreiter et al., 2001*). These conditions are unlikely to be met or maintained in the environment of temporary pools. Although the degeneration of DNA in water is greatly accelerated, small fragments can remain detectable for up to a month (*Dejean et al., 2011*). However, intracellular DNA, like that within a degraded egg, could be somewhat more protected from abiotic and biotic factors and degenerate at a slower rate (*Nielsen et al., 2007*). The identification and counts of these degraded eggs could be used as a proxy for the overall condition of an egg bank, presenting a view of recent mortality rates in the diapausing eggs.

Here we developed species-specific DNA barcoding of isolated eggs to identify viable *Triops cancriformis* diapausing eggs from sediments. We compared the results obtained

with this method with two conventional alternatives: sediment hatching and isolation hatching over two hydroperiods to account for bet-hedging. We applied the three methods to 12 sediment samples collected from ephemeral pools at the WWT Caerlaverock Wetland Reserve, including pools where *Triops* had been previously recorded plus some potential new sites. We estimate diapausing egg bank size, egg viability and condition in these pools. In addition, from the collected mtDNA data we also describe the genetic diversity of the Caerlaverock populations in the context of available data from other European populations. Our method could be used as a time efficient strategy for discovering and monitoring the viability and health of *T. cancriformis* egg banks across Europe.

## MATERIALS AND METHODS

All work was carried out under Scottish Natural Heritage licence number 42854.

### Sample collection and preparation

We sampled 12 temporary pool sites on the WWT Caerlaverock reserve from the 10th to the 11th of September 2015. Eight sites, including the site of the species rediscovery in 2004, were located on the Eastpark Farm holding of the reserve along the cattle grazed scrub and grassland bordering the Solway Firth estuary mudflats. The other four sites were on cattle grazed pasture on the Powhillon Farm holding on the north of the reserve. Sites consisted of either temporary pools where *Triops* had been recorded before (either through presence of *Triops* or where past experiments yielded *Triops* hatchings) or sites with no previous *Triops* records but apparently suitable *Triops* habitat in that they had regular hydroperiods and had been recorded to dry out at least once a year.

At each site GPS coordinates were obtained from the centre of the pool using an eTrex Camo GPS device (Garmin Ltd, Olathe, KS, USA). Using a stainless steel spoon around 500 g of superficial sediment (ca. top 2.5 cm) was collected from eight uniformly distributed sample points, four around the pool centre and four midway to the pool boundary. Sampling spoons were thoroughly cleaned of all sediment and debris after each site to avoid cross-site contamination. Collected sediment from a site was placed directly into large labelled Ziploc bags, which were immediately placed into another identical bag to further prevent cross-site contamination. Once in the laboratory, collected sediment samples were placed in separate open topped 2 L plastic jars and left to dry out over a period of four weeks at 20 °C. Once completely dry the samples were gently crumbled into a finer state by hand. Three subsamples of 20 g were then taken from each sample to be used in sediment hatching, isolation hatching and DNA barcoding, respectively.

### Comparison of methods

DNA barcoding of isolated *T. cancriformis* diapausing eggs was compared to conventional survey methods of sediment hatching and isolation hatching. Each method gave an estimate of viable eggs per site. Total egg counts were achieved via diapausing egg isolation from sediment. Viable and total egg counts were recorded from all three methods per site and compared to evaluate our molecular approach (Fig. 1). These counts also allowed for the calculation of proportion viability and egg bank density (eggs/kg) per site. Additionally

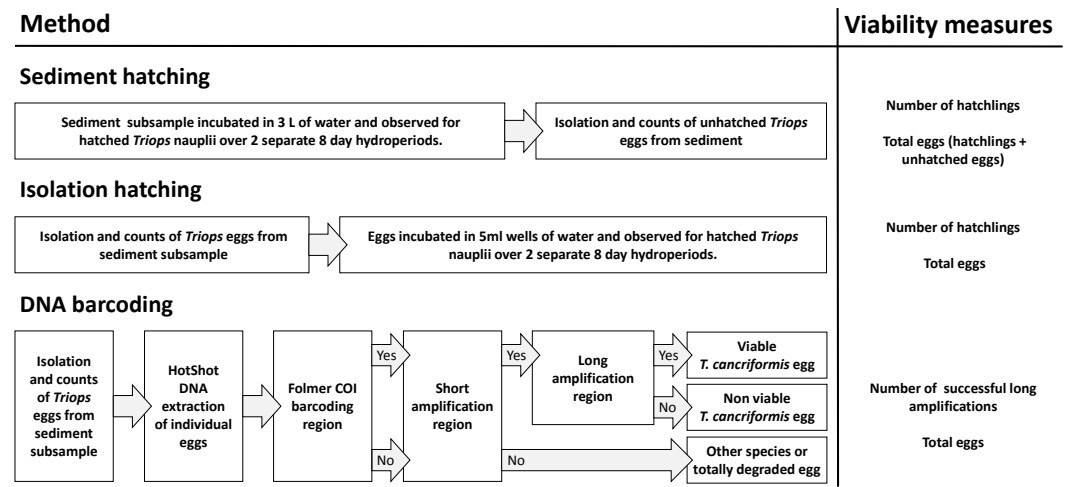

**Figure 1** **An overview of the three methods used in this study: sediment hatching, isolation hatching and DNA barcoding.** Shown are the simplified steps undertaken for each procedure. "Yes" and "No" in the DNA barcoding flow chart refers to successful and unsuccessful amplifications, respectively. The viability measures obtained for each site were used for comparison to determine the value of our molecular approach and to calculate proportion viability and egg bank density (eggs/kg) per site.

unhatched eggs from the sediment and isolation hatching methods were tested for viability with the DNA barcoding method. Estimates of the time and costs involved in each method were also compared.

## Isolation of diapausing eggs from sediment

Eggs were isolated from sediment in all of the three methods of this study as a means for DNA extraction of individual eggs for DNA barcoding, actual isolation of eggs for isolation hatching and the counting of remaining unhatched eggs after sediment hatching experiments had concluded. Identification of *T. cancriformis* eggs was achieved through comparison to known example specimens and collections within the laboratory and to those within the literature (*Kuller & Gasith, 1996*). Diapausing *T. cancriformis* egg isolation from collected sediment samples followed a sucrose flotation method adapted from *Gómez & Carvalho (2000)*. In short, 5 g of sediment was added to 50 mL sucrose solution (50/50 w/v sugar/water). This was thoroughly mixed by vortexing before being centrifuged at 700 rpm. Resting eggs were then captured from the supernatant with a 50 μm Nytal filter. Washed and rehydrated filtrate was observed under a stereoscopic microscope for *T. cancriformis* eggs. Identified *T. cancriformis* eggs were collected using a 200 μL Gilson pipette and transferred to a sterile small welled cell culture plate (Corning Costar 3526, 24 well, flat bottomed culture plate; Thermo Fisher Scientific, Waltham, MA, USA). Isolated eggs were further checked under a stereoscopic microscope to determine if they were hatched or unhatched. Hatched eggs were discarded. The number of unhatched eggs isolated was recorded for each subsample.

## Hatching experiments

Hatching experiments took place from the 16th of November to 19th of December 2015 in conditions optimised for *Triops* hatching: incubated in purified water in a temperature controlled growth room at ∼20 °C (*Eder, Hödl & Gottwald, 1997*; *Schönbrunner & Eder, 2006*) with a 12/12 day/night cycle (*Kuller & Gasith, 1996*; *Kashiyama et al., 2010*) under white fluorescent light tubes. To account for possible bet-hedging strategies of *T. cancriformis*, hatching was undertaken in two short hydroperiods of eight days, each with a seven day drying period in between. To make best use of available space, samples were run in batches of six, A–F and G–L, staggered weekly such that as one batch was drying the other was undergoing a hydroperiod. Both sediment and isolation hatching for each site were run alongside one another so as to be under the same lighting and temperature variations over the observation periods.

### Sediment hatching

A 20 g subsample of sediment from each site was added to a 6 L lid-less acrylic tank ($L = 30$ cm, $D = 20$ cm, $H = 20$ cm) filled with 3 L of purified water. The sediment was allowed to settle for an hour before being evenly distributed across the tank bottom using a large spatula. The water of each tank was gently agitated with a constantly running air pump for the duration of the experiments. Hatchlings were searched for in each tank daily for a period of about ten minutes. Any hatched *Triops* nauplii were removed using a 1,000 µL Gilson pipette, placed in a separate Petri dish for each site and counts recorded. Removed nauplii were preserved in 70% ethanol or underwent HotShot DNA extraction (*Montero-Pau, Gómez & Muñoz, 2008*) for future use. Observations were carried out over an eight day period. On the last day after observations the tanks were drained using a thin tube to siphon the water through a 50 µm Nytal filter device. A separate filter was used for each tank. The filters were then examined under a stereoscopic microscope and any hatchlings present were added to the recorded count for the day and preserved. Any eggs found were placed back in the corresponding tank sediment. The sediment was left in the tanks to completely dry out over a period of seven days before being refilled with 3 L of purified water and the above process repeated for the second hydroperiod.

After the drying period following the second hydroperiod was completed the sediment was removed from the tank. Any remaining eggs were isolated from the sediment using the sucrose flotation method described above. Unhatched egg numbers were recorded and added to the total number of hatchlings for each site as a proxy for the total number of initial eggs present in the subsample.

### Isolation hatching

Eggs were isolated from a 20 g subsample of the dried sediment from each site using the sucrose flotation method as described above. Immediately after being isolated, eggs from each site were placed in 1.5 mL of purified water in a sterile cell culture plate (Corning Costar 3526, 24 well, flat bottomed culture plate; Thermo Fisher Scientific, Waltham, MA, USA), in groups of up to five per cell. The plate cover was placed on top to reduce evaporation. Over the following eight day period ten minute observations of each plate were performed daily and any hatched *Triops* nauplii were removed using a 1,000 µL

Gilson pipette, placed in a separate Petri dish for each site and counts recorded. Removed nauplii were preserved in 70% ethanol or underwent HotShot DNA extraction for future use. On the eighth day after observations the wells were carefully drained using a 1,000 μL Gilson pipette. The plate wells were left with the covers removed to completely dry out over a period of seven days before being refilled with 1.5 mL of purified water and the above process repeated for the second hydroperiod.

## DNA barcoding

The molecular method of DNA barcoding of isolated *T. cancriformis* eggs was designed to produce simple PCR steps to identify viable eggs and the diapausing egg bank condition (Fig. 1). Two species-specific primer pairs were designed for this study. A primer pair to amplify a large 2,500 bp target region of mtDNA (long amplification) so that presumably only intact, viable *Triops* eggs amplified. A second primer pair to amplify a short 132 bp target region of mtDNA (short amplification) to act as a species identifier for degraded DNA, potentially found in *T. cancriformis* inviable eggs. DNA extraction samples from all individual isolated eggs were first amplified with the universal DNA barcoding primers LCO1490 and HCO2198 (*Folmer et al., 1994*) to give a ∼650 bp fragment of cytochrome c oxidase subunit I gene (COI). This first step was aimed at determining the taxonomic identity of all samples that failed to be identified as *T. cancriformis* via subsequent short and long amplifications. All samples then underwent PCR with the short amplification primers, identifying which samples had *T. cancriformis* mtDNA present. Those samples with successful COI amplifications underwent PCR with the long amplification primers. This step would confirm both the designed primers of this study to be species-specific, as any sample with a long amplification should have a complementary short amplification. Samples with a successful short amplification and no long amplification present were considered to be degraded *T. cancriformis* eggs.

Finally, after the completion of the second hydroperiods for both sediment and isolation hatching, all remaining unhatched eggs were removed and underwent the DNA extraction protocol (see details below) and underwent PCR for the long amplification region. To check if all viable eggs had hatched in our hatching experiments and to confirm the suitability of long amplifications to identify viable eggs, all DNA extractions from unhatched eggs from the hatching methods were amplified for the long amplification region.

### DNA extraction of isolated eggs

*T. cancriformis* eggs from a 20 g sediment subsample were isolated as described above. Genomic DNA was individually extracted using the HotShot DNA extraction protocol from *Montero-Pau, Gómez & Muñoz (2008)*. A total of 50 μL of lysis buffer was aliquoted into 0.2 mL Eppendorf tubes. A single isolated *T. cancriformis* egg was transferred into each tube using a 200 μL Gilson pipette. The egg was crushed on the side of the tube within the lysis buffer with a sterile 10 μL Gilson pipette tip. Tubes were incubated at 95 °C for 30 min followed by cooling on ice for 5 min. 50 μL of neutralising solution was then added to each tube then vortexed and centrifuged. All HotShot extractions were stored at −20 °C until required.

**Table 1  Primers designed and developed in this study.** Primer sequences and product size are given.

| Primer pair name | Primers | Primer sequences (5′–3′) | Product size (bp) |
|---|---|---|---|
| Long amplification | GS-Tyr-1349F | AGGGGAAACTCCCATATTTAGATT | 2,500 |
| | GS-ATP8-3806R | TACTAGGGGCTATTTGGGGG | |
| Short amplification | GS-trnaS-5881F | TGCATTCAAAAGGTACTACCAAAA | 132 |
| | GS-trnaS-5971R | TGCCGATCATTGGCTTCAA | |

### *Primer design*

Species-specific primers were designed and tested *in silico* with Primer BLAST (*Ye et al., 2012*) using the complete *T. cancriformis* mitochondrial genome as a reference sequence (Genbank accession number AB084514.1) (Table 1). The long amplification region was located from tRNA$_{Tyr}$ to ATP8. This region encompassed the whole Folmer COI region for comparison to existing *T. cancriformis* COI sequences. The short amplification region was located across tRNA$_{Ala}$ and tRNA$_{Asn}$ after the ND3 gene.

All primers were tested *in vitro* on three species of *Triops* (*T. cancriformis*: Caerlaverock, Scotland; Espolla, Spain; Königswartha, Germany, *T. mauritanicus*: Doñana, Spain and *T. newberryi*: Triop World (Interplay UK; Marlow, Buckinghamshire, UK)) and several freshwater invertebrate specimens from Caerlaverock and other UK locations (*Daphnia sp.*, Ostracoda and Copepoda). DNA templates were from HotShot DNA extractions of hatched specimens and collected tissue samples. The long amplification primer pair amplified only *T. cancriformis* and its sister species *T. mauritanicus*. The short amplification primer pair was found to be completely specific to the target species. PCR cycling conditions were optimised for both long and short amplification primer pairs.

### *PCR amplification*

All PCRs were performed on Applied Biosystems Veriti 96-Well Thermal Cyclers in a 25 μL final reaction volume composed of 2 μL template DNA, 12.5 μL MyTaq$^{tm}$ Red Mix (Bioline, London, UK), 8.5 μL ddH$_2$O and 1 μL of each 10 μM primer. PCR products were visualised on 1.5% agarose gels. COI PCRs were run under the cycling conditions: 180 s at 94 °C, 37 × (30 s at 94°, 60 s at 52 °C, 90 s at 72 °C), 600 s at 72 °C. Short amplification PCRs were run using the designed primers GS-trnaS-5881F and GS-trnaS-5971R, under the cycling conditions: 180 s at 94 °C, 37 × (30 s at 94°, 30 s at 55 °C, 30 s at 72 °C), 600 s at 72 °C. Long amplification PCRs were run using the designed primers GS-Tyr-1349F and GS-ATP8-3806R, under the touchdown cycling conditions: 180 s at 94 °C, 10 × (30 s at 94 °C, 60 s at 70 °C [−1 °C per cycle], 105 s at 72 °C), 27 × (30 s at 94 °C, 60 s at 60 °C, 105 s at 72 °C), 600 s at 72 °C. Faint amplifications were rerun with a 1:20 template dilution to reduce any PCR inhibition or DNA overloading. Positive (previously successful *T. cancriformis* nauplii extractions) and negative controls were used in each PCR batch.

## DNA sequencing

To confirm the the specificity of our designed primers, PCR products from five samples with successful short amplifications and five with successful long amplifications were sequenced. To discover possible discrepancies over the COI region, a further 20 samples with successful

COI and long amplifications had both PCR products sequenced. Finally, in order to validate our molecular diagnostic tool and to verify the identity of eggs and identify those that could be confused with *Triops*, all samples not identified as *T. cancriformis* via a long amplification that had successful COI fragments were sequenced. COI and long amplification products were sequenced using the LCO1490 primer. Short amplification products were sequenced using the GS-trnaS-5881F primer. All sequencing was performed by Macrogen (Seoul, South Korea). Sequences were manually edited using CodonCode Aligner (CodonCode Corp.; Dedham, MA, USA). End clips were performed to remove low quality regions from both ends of the sequences (end regions containing more than 3 bases with lower quality than 20 within a 25 bp window were trimmed). Sequences shorter than 50 bp after clipping were discarded as poor quality. Furthermore, samples with COI and long amplification fragments shorter than 100 bp were also discarded as poor quality for this size region is unreliable for successful COI identification (*Meusnier et al., 2008*). Remaining sequences were put through NCBI BLASTn for sequence identification. All good quality *T. cancriformis* COI and long amplification sequences were submitted to GenBank (accession numbers: KY769474–KY769517).

## mtDNA population network

A population network was created to compare the COI haplotypes from the WWT Caerlaverock population to other *T. cancriformis* sequences across Europe. Sequences in this study identified as *T. cancriformis*, from either a COI or long amplification of an individual sample were aligned to all *T. cancriformis* COI sequences available from Genbank. *T. mauritanicus* was used as an outgroup. Sequences were aligned and trimmed to 512 bp using Aliview (*Larsson, 2014*) and any shorter sequences were discarded. POPART (http://popart.otago.ac.nz.) was used to create a TCS statistical parsimony network (*Clement et al., 2002*).

## Egg bank density, viability and condition

Egg bank density was estimated to measure the number of eggs per kg sediment in a site. Proportion viability was estimated to measure the overall viability of the egg bank in a site and was the primary measurement used for the statistical comparison of the three methods. Viable egg counts from all three methods per subsample were used to calculate a proportion viability for each method per site (see Fig. 1). For sediment and isolation hatching, the number of viable eggs was estimated as the total number of hatchlings over two hydroperiods in a site subsample. For DNA barcoding the number of viable eggs in a site subsample was estimated as the number of successful long amplifications. Estimated egg bank density for each site was calculated from the average total egg counts per site from the three methods.

Using the barcoding method described here, the condition of an egg bank can be inferred through the proportions of viable (samples with long amplification), degraded (samples with a short amplification and no long amplification) and totally degraded eggs (samples with neither long nor short amplifications) present. For each site the proportion of viable, degraded and totally degraded eggs were calculated and combined with egg bank density (eggs/kg sediment) to present a measure of egg bank condition. Egg bank density was estimated from the total number of eggs isolated per site from DNA barcoding.

## Statistical analyses

We tested for statistical differences between the estimated viability per site across the three tested methods (isolation hatching, sediment hatching, DNA barcoding). We used a general linear mixed model of viable against nonviable egg counts with binomial errors implemented in R (*R Core Team, 2013*) (version 3.2.5, package "lme4" *(Bates et al., 2015)*). "Site" was used as a random variable and "method" as a fixed variable. To determine if method was a significant factor in any variances in measures of egg viability, we compared this model to the same model with no fixed variable using a chi-squared test of the likelihood of models. Overdispersion was tested for in both models (R version 3.2.5, package "blmeco" (*Korner-Nievergelt et al., 2015*)).

# RESULTS

## Sample collection

Sample sites were located in grazing pasture with cattle present. Sites D, E and F were located in wheel ruts along tractor trails linking grazing pastures. At the time of sampling sites B, C, H and L had water up to a depth of 10 cm remaining. Sites A, B, C, I and L had sparse vegetation growth within the pool boundaries. All other sites were dry, or drying, exposed sediment. Two weeks prior to sampling a tidal surge up the Lochar Water, a river that runs through the reserve, had breached its small defence walls and flooded the eastern side of the Powhillon Farm field area that included sample sites J and K. At the time of sample collection water samples taken from a remaining large pool, adjacent to sites J and K, had a salinity of 17.5 ppt. Estuarine and marine species were found alive within these pools or exposed on the drying pool sediments, *Crangon crangon* (brown shrimp) in drying sediment at site J, *Pungitius pungitius* (ninespine stickleback) within the large saline pool, and juveniles of *Carcinus maenas* (green shore crab) at site K. During sampling there was evidence of *Triops* presence in one of our sampled sites (site K) where no records existed before, with many exuviae present in caked sediment.

## Isolation of resting eggs from sediment

*Triops cancriformis* eggs were isolated from all sites, therefore all sampled sites held a *T. cancriformis* egg bank of varying density. Two sites had distinctly larger egg banks than the other sites sampled: site G, the site of *T. cancriformis* rediscovery at the WWT Caerlaverock Wetland Reserve in 2004, and site J on the Powhillon Farm holding of the reserve. All identified *T. cancriformis* eggs were not in similar condition: many having begun to lose the external coating of fine sediment particles or appearing flat and misshapen.

## Hatching experiments

Six of the 12 sample sites produced *T. cancriformis* nauplii from sediment and isolation hatching methods, however not all sites exhibited hatchlings from both methods (Table S1). Sites G and J had the highest hatching rates. Site K, with no previous records of *T. cancriformis* presence, had a single recorded isolation hatchling. Site E had previous records of *T. cancriformis* presence but had no hatchlings recorded from either hatching method.

**Table 2  Outcome from *T. cancriformis* diapausing egg DNA barcoding.** Counts of samples with COI, long and short amplification PCR combinations and total eggs processed for each site are given. The six sites with recorded hatchlings from sediment and or isolation hatching are marked with asterisks.

| PCR combination | | | Site | | | | | | | | | | | |
|---|---|---|---|---|---|---|---|---|---|---|---|---|---|---|
| COI | Long | Short | A | B | C | D* | E | F* | G* | H | I* | J* | K* | L |
| ✓ | ✓ | ✓ | – | – | – | 1 | – | 1 | 25 | – | 3 | 22 | 2 | – |
| ✓ | – | ✓ | – | – | – | – | – | – | 5 | – | – | 2 | 5 | – |
| – | – | ✓ | – | – | – | – | – | – | – | – | 1 | – | – | – |
| ✓ | – | – | 2 | 1 | 4 | 1 | 2 | 9 | 25 | 6 | 6 | 22 | 5 | 3 |
| Total eggs | | | 18 | 11 | 23 | 5 | 6 | 10 | 60 | 6 | 17 | 51 | 13 | 6 |

Just four nauplii hatched in the second hydroperiod of the hatching experiments overall, one in site G sediment hatching and three in site J isolation hatching, indicating low bet-hedging strategies in these populations.

Over the first hydroperiod, hatched nauplii from both hatching methods were recorded within a small time window over the eight day observation period (Fig. S1). The first hatchlings were recorded after a 48 h incubation period. Most hatchlings appeared on days two to five across both methods. As sediment hatchlings were more difficult to spot compared to those of isolation hatching, a small number of sediment hatchlings may have been overlooked and only discovered on later observation days than those of isolation.

## DNA barcoding of isolated eggs

A total of 226 individual eggs were processed using DNA barcoding, of which 153 yielded positive amplifications with at least one of the primer pairs (Table 2). Samples from all sites yielded positive Folmer COI region PCR amplifications yet those with positive long and short amplifications were only present in the six sites with recorded nauplii in hatching experiments: sites D, F, G, I, J and K (Table 2). Short amplifications were associated with samples that had a successful COI amplification with the exception of a single sample from site I. All samples with successful long amplifications also had successful short amplifications.

### DNA sequencing

All five short amplification and four of the five long amplification sample sequences were all good quality and identified as *T. cancriformis*, confirming the specificity of our designed primers (Table S2). The 20 samples with both COI and long amplification products all had good quality COI sequences that were identified as *T. cancriformis* (Table S3). Of the long amplification sequences, eight were of poor quality and discarded. The remaining 12 were of good quality and all identified as *T. cancriformis*. There were no discrepancies between results as the Folmer COI region from long amplification and COI sequences were identical across all samples, further confirming the suitability of the long amplification for species identification. Out of the 97 samples with COI amplifications (with no long amplification) sequenced 41 were of poor quality and discarded. The top hits of the NCBI BLASTn returns for the remaining samples showed eight *T. cancriformis* sequences and 48 non-*Triops* sequences (Table S4). All the *T. cancriformis* COI sequences were from five of the six sample sites with recorded hatchlings from this study (sites D, G, I, J and K). All non-*Triops*
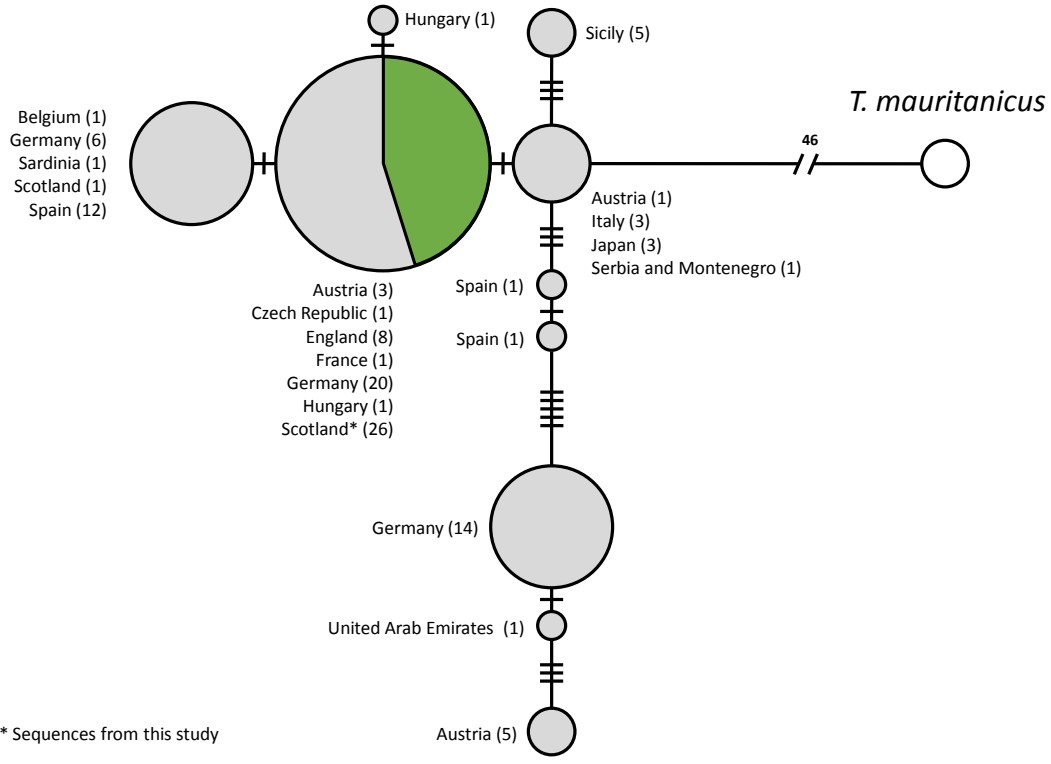

**Figure 2  Statistical parsimony network of COI sequences from _T. cancriformis_ isolates with _T. mauritanicus_ as an outgroup.** Those in green are from this study. Countries of origin and the number of isolates are given next to each node. Ticks on linkages indicate number of mutations between nodes.

sequences were identified to species with no similar egg morphology to _Triops_. Two samples from site G had short amplifications with no long amplification and a non _Triops_ COI sequence identified. Three samples, one from site G, I and K, had a short amplification with a non-existent or poor quality COI sequence.

## Population mtDNA network

A total of 115 COI sequences of individual _T. cancriformis_ isolates, including 26 from this study, with one _T. mauritanicus_ isolate as an outgroup were used to produce the TCS mtDNA haplotype network (Accession numbers and sample ID in Table S5). We found a single COI haplotype in our Caerlaverock _T. cancriformis_, which is identical to a common haplotype found in a large number of isolates from Europe, including isolates from the other British population in the New Forest (Fig. 2). Intriguingly, the only previously analysed Scottish sample contained a haplotype differing from those of this study in one base pair.

## Comparison of methods for determining egg bank viability

Estimates of proportion viability of egg banks varied between sites, with only 6 out of the 12 sites showing viable egg banks, with the maximum viability found in site G (Fig. 3). DNA barcoding was the most powerful method to detect sites with viable eggs (six sites) compared to isolation hatching (five sites) and sediment hatching (four sites). The three methods gave similar results in sites with larger more uniform egg counts: sites G, I and

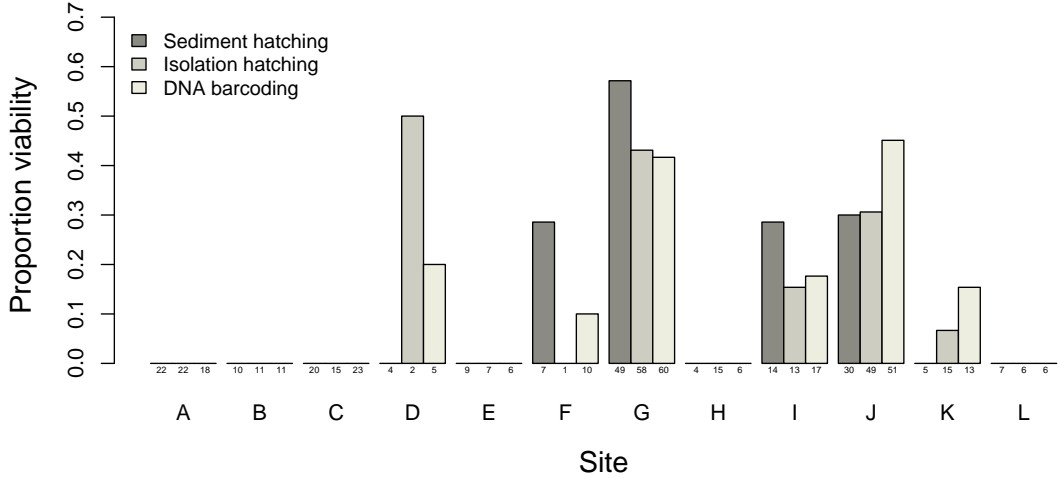

**Figure 3** **Proportion of *T. cancriformis* viable eggs per site from the three methods employed (sediment hatching, isolation hatching and DNA barcoding each from a 20 g subsample).** Total number of eggs per method are listed under the columns.

J. There was no significant difference in egg viability estimates between the three methods across all sites ($X^2 = 1.7995$, $df = 2$, $p = 0.4067$) with no overdispersion in either model. Therefore, successful DNA barcoding of long amplifications can be used as a reliable measure of viability in *T. cancriformis* resting eggs.

### DNA barcoding of unhatched eggs

DNA barcoding using the long amplification primers on extractions from the unhatched eggs remaining after sediment and isolation hatching experiments was in general unsuccessful. Only 12 eggs out of 308 samples had successful long amplifications (Table S6). These were in sites with the larger sample sizes (G and J) suggesting some bet-hedging in these populations that the hatching methods failed to detect over the two hydroperiods. Site J isolation hatching showed the highest number of unhatched eggs amplifying the long amplification primers, with eight identified. As we wanted to use long amplification as a proxy for viability, and to determine if there would have been any effect upon the estimated viability between methods had these eggs hatched during the experiments, the GLM analysis was rerun with adjusted results. There was again no significant difference between the three methods used to determine viability ($X^2 = 0.6954$, $df = 2$, $p = 0.7063$) with no overdispersion in either model.

### Comparison of time expenditure and costs

Given the budget constraints of environmental monitoring, we estimated time expenditure and equipment cost for each method to produce an egg bank viability estimate based upon a single high egg count subsample (60 eggs per subsample) (Table S7). Drying of collected sediment was not factored into the comparison. The methods were divided into processes. Each process was evaluated by the time to its completion and the maximum time a researcher would have to expend executing it. Times for hatching setups were ignored as they were either part of a previous procedure, as for isolation hatching, or considered

negligible (less than 2 min), as for sediment hatching. The PCR time was calculated for running a single 60 sample PCR preparation and amplification using long amplification primers. Consumables costs were based upon approximate retail values of materials used that could not feasibly be reused for the same process. Salary times were not costed, just time expenditure calculated.

Both sediment and isolation hatching take several weeks to complete (over 32 days and 24 days respectively), considerably longer than the DNA barcoding method to achieve the same result (7.5 h). Although all three methods require a similar input of time to process (around 4 h of a researcher's time), this is spread over a much greater time frame for both hatching methods than for DNA barcoding. In contrast to time efficiency, consumables costs for the hatching methods are a minimal amount (0.20 GBP) compared to those of DNA barcoding (30.00 GBP).

### Egg bank density, viability and condition of *Triops cancriformis* populations at Caerlaverock

All sites had an egg bank present based upon calculations from the three methods, yet egg bank density (eggs/kg sediment) estimates varied between sites. Two sites had higher densities: sites G and J (Table S8). Viable egg banks are found across the reserve but are clustered around the two higher density sites G and J (Fig. 4). From the molecular method the proportion of viable eggs (long amplifications), degraded eggs (short amplifications) and totally degraded eggs (those with neither amplifications) were combined with egg bank size to give a representation of overall condition (Fig. 4, Table S9). The 12 sites had differing proportions of viable, degraded and totally degraded eggs. Overall, there was a high proportion of totally degraded eggs but six of the sites had egg banks in a totally degraded condition, that is, non-existent (sites A, B, C, E, H and L). In contrast, sites G and J had high proportions of viable eggs. Sites D and F had no degraded eggs present. Sites G, I and J showed low proportions of degraded eggs in comparison to that of viable eggs. Site K had a much higher proportion of degraded eggs than that of viable eggs. Sites with high proportions of viable eggs and low proportions of degraded eggs were interpreted as having *T. cancriformis* egg banks in good condition. Sites with higher proportions of degraded eggs than viable were of poor condition. Sites with no viable *T. cancriformis* eggs, might have held populations in the past, but the species can be considered to have become extinct. All but one of the sites with viable *T. cancriformis* egg banks determined from this study had recent records of the species: site K. In this site we found exuviae at the time of sampling and it is a new location for the species. In site E *Triops* adults were recorded in 2013, however we failed to find viable eggs from all methods employed, suggesting that this population might have become extinct.

## DISCUSSION

This study describes a powerful and efficient molecular technique that can identify viable *T. cancriformis* eggs isolated from sediment samples outperforming conventional incubation methods, therefore helping to discover new populations and monitor existing ones. Primarily, a single PCR using species-specific long amplification primers on DNA extracted from

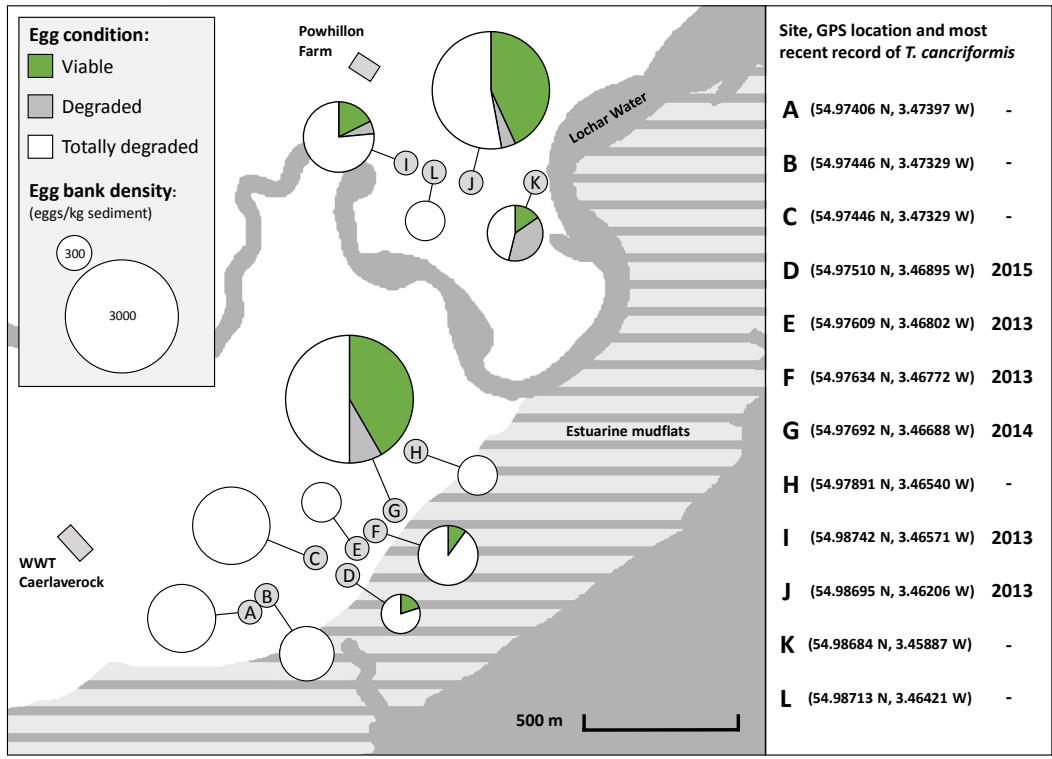

**Figure 4  Location and condition of *T. cancriformis* egg banks sampled across the WWT Caerlaverock Wetland Reserve.** Shown are proportions of viable, degraded and totally degraded eggs per site as determined by our molecular method. Chart size proportional to estimated egg bank density (eggs/kg sediment) as determined by the molecular method. The GPS location and year of the most recent recorded *T. cancriformis* presence (adults or hatchlings) for the sites are shown for the period up to the time of sampling in September 2015 (right).

eggs isolated from a sediment sample gave an estimate of viable eggs present. Secondarily, a further PCR using the short amplification primers on the same DNA extractions confirmed the species as *T. cancriformis* and could be used to estimate the number of degraded eggs present: those with no successful long amplification. The combination of these results with the total number of isolated eggs from sediment samples provided an overview of egg bank condition. All good quality long amplification sequences were identified as *T. cancriformis* and the viability estimates obtained from the molecular approach were not statistically different from the sediment and isolation methods across all sites. Most eggs that remained unhatched in both hatching experiments after two rounds of hydration failed to amplify with the long amplification primers, validating the use of our molecular technique to estimate diapausing egg bank viability. However, the fact that a few of these eggs did amplify suggested the presence of a certain amount of bet-hedging in these *T. cancriformis* populations. These were not included in the hatching methods viability measures so reduced the estimates of viability for the sites. This means that our molecular method produces a viability measure for *T. cancriformis* diapausing egg banks, removing any uncertainty of bet-hedging for a complete viability estimate.

The molecular method, as with the hatching methods, relied upon initial morphological egg identification from samples. Our visual identification of *T. cancriformis* was confirmed via the COI DNA barcoding of samples, with most good quality COI and long amplification sequences belonging to *T. cancriformis*. Other good quality COI sequences obtained did not include groups with diapausing egg morphology similar to *T. cancriformis*. Non *T. cancriformis* COI sequences were mostly of bacteria, microalgae and water moulds associated with ephemeral pools that inhabited, were adhered to or present within the sediment attached to a degraded egg (Table S4). Our data also show that environmental DNA from larger organisms found in and around the habitat pervaded the sample.

Both hatching methods showed a similar pattern of emergence and numbers of hatchlings. Therefore sucrose flotation of *Triops* eggs used in the isolation hatching method had no effect upon hatching rates of resting eggs, as recently supported by *Lukic, Vad & Horváth (2016)*. Our results from the hatching methods suggest that the Caerlaverock *T. cancriformis* populations exhibit a low level of bet-hedging. This is further supported by the few successful long amplifications found in the remaining unhatched eggs of the hatching methods.

Previously, the estimated condition of a species' diapausing egg bank had only been achieved with rotifers via visual inspection of individual egg appearance (*García-Roger, Carmona & Serra, 2005*). Unlike the conventional survey methods used for *T. cancriformis* monitoring, the molecular method used in the current study can similarly estimate the condition of a *T. cancriformis* egg bank through the identification of viable, degraded and totally degraded eggs. Egg banks in the six sites with high proportions of viable eggs (samples with long amplifications) can be considered to hold good condition, viable *T. cancriformis* populations. Mortality rates within an egg bank can be inferred from the proportion of degraded eggs (samples with only short amplifications) present. As these eggs have relatively recently deteriorated it can be used as a proxy for mortality events from external factors, be they biotic or abiotic. Some sites had small sample sizes due to low egg bank densities and would require larger sample sizes to get better representations of condition. In contrast, the remaining six sites, with only totally degraded eggs, do not currently hold *T. cancriformis* populations. During this study we discovered a new population of *T. cancriformis* on the WWT Caerlaverock reserve (site K) and also determined that a previously recorded population (site E) might now have become extinct. This suggests a certain degree of dynamism in population persistence, potentially reflecting the existence of dynamic metapopulations in the area, as it is the case of other temporary pool branchiopods such as *Daphnia* (*Ebert et al., 2002*; *Haag et al., 2005*).

We used the sequences obtained to validate our methods by comparing Caerlaverock samples to other *T. cancriformis* populations. Our analysis showed that Caerlaverock mtDNA belongs to the most common European COI haplotype of *T. cancriformis* (Fig. 4). The fact that the only previously sequenced Caerlaverock sample from *Zierold, Hänfling & Gómez (2007)*, belonging to a different haplotype, could potentially reflect diversity not sampled in our study.

The molecular method of this study is a more efficient method for determining the presence of a viable *T. cancriformis* egg bank than the conventional and standardised methods of sediment and isolation hatching. Additionally with the use of species-specific primers

the cost of sequencing is removed, both in terms of time and money, setting our method apart from other molecular approaches that rely upon sample sequencing to determine species identity. A successful amplification viewed via gel electrophoresis can be used to confidently identify the organism as *T. cancriformis* and, as with the long amplification, the viability of a resting egg. As a direct comparison of time frames involved in this study the molecular analysis of a single site, from egg isolation to gel electrophoresis, took a matter of hours (Table S7), whereas the hatching experiments took over three weeks to complete (four in the case of sediment hatching). When dealing with much greater sample sizes, as with *Adams et al. (2014)*, the time expenditure can be greatly reduced using our molecular method. The major drawback to the method is the consumables cost. With the hatching methods the only consumable was the sugar used in the sucrose flotation method to isolate the diapausing eggs. This is distinctly inexpensive when compared to the consumable costs for the molecular method which were many times greater than those of the hatching methods (Table S7). Salary costs were not included as the staff time for each method was very similar. However during the extended time frame of the hatching methods there are periods of daily observations to be undertaken requiring a researcher's presence, which would increase the overall economic costing of the hatching methods.

Molecular approaches, in particular eDNA, are increasingly used to determine the presence of endangered species in freshwater habitats, as with the Great Crested Newt, *Triturus cristatus*, in the UK (*Rees, Bishop & Middleditch, 2014*) and multiple species in Europe (*Thomsen et al., 2012*), and can detect secretive or rare species more effectively than conventional methods (*Hänfling et al., 2016*; *Valentini et al., 2016*). The molecular method presented in this study not only efficiently detects viable *T. cancriformis* populations, directly addressing the needs for heightened surveillance for *T. cancriformis* populations as raised by *Adams et al. (2014)*, but provides better estimates of egg bank density and condition. Our methods have conservation implications not only for British *T. cancriformis* populations, but more widely as they were designed and tested on European populations. The implementation of an effective method for determining the presence and condition of viable *T. cancriformis* populations across the species' distribution reduces the time costs considerably. The ease of processing many samples, with bet-hedging uncertainties removed, will give accurate, reliable and rapid results for implementation of relevant conservation measures. Additionally our molecular method can be used for the sister species of *T. cancriformis*: *T. mauritanicus*, meaning that the diagnostic tools presented here would be useful for the monitoring of viable *T. cancriformis* and *T. mauritanicus* populations.

## CONCLUSION

The potential accelerated loss of habitat suitable for endangered *T. cancriformis* populations across the species distribution requires an effective survey method for its conservation. We present a powerful alternative molecular method that, through the amplification of mtDNA extracted from isolated eggs using species-specific primers, can reliably and efficiently determine the presence, condition and viability of *T. cancriformis* egg banks. The complications

of passive dispersal, extended diapause and bet-hedging are removed as, unlike conventional survey techniques, our method does not rely on observations of hatched or adult individuals to discover an extant *T. cancriformis* population. The increasing success and decreasing cost of molecular techniques for ecological conservation and diversity monitoring (*Thomsen et al., 2012*; *Lawson Handley, 2015*) make them viable alternative approaches. The use of designed species-specific primers alleviates the cost of sequencing, further reducing the costs. Implementation of our molecular method will present a cost-effective and efficient tool for the discovery and monitoring of *T. cancriformis* populations in the UK and Europe. From the results of this study, the current management of WWT Caerlaverock is ideal for maintaining the dynamic metapopulation of *T. cancriformis* that appears to be present across the reserve.

## ACKNOWLEDGEMENTS

We would like to thank Dr. James Kitson and Dr Paul Nichols for advice regarding molecular protocols, Dr. Amir Szitenberg for guidance with molecular data analysis tools, Dr. James Gilbert for statistical assistance, Dr. Lori Lawson Handley and Dr. Lesley Morrell for advice and feedback. We thank the Wildfowl & Wetlands Trust for support during the sampling trip to Caerlaverock.

### Funding

This research was funded by a PhD scholarship from the University of Hull awarded to Graham S. Sellers. The funders had no role in study design, data collection and analysis, decision to publish, or preparation of the manuscript.

### Grant Disclosures

The following grant information was disclosed by the authors:
PhD scholarship from the University of Hull.

### Competing Interests

The authors declare there are no competing interests.

### Author Contributions

- Graham S. Sellers conceived and designed the experiments, performed the experiments, analyzed the data, contributed reagents/materials/analysis tools, wrote the paper, prepared figures and/or tables, reviewed drafts of the paper.
- Larry R. Griffin, Bernd Hänfling and Africa Gómez conceived and designed the experiments, wrote the paper, reviewed drafts of the paper.

### Field Study Permissions

The following information was supplied relating to field study approvals (i.e., approving body and any reference numbers):

All field sampling was performed under Scottish Natural Heritage licence number 42854.

## DNA Deposition

The following information was supplied regarding the deposition of DNA sequences:

All *Triops cancriformis* mitochondrial cytochrome c oxidase I sequences from this study were submitted to GenBank (accession numbers KY769474–KY769517).

## Data Availability

The raw data has been supplied as Supplementary Files.

## Supplemental Information

Supplemental information for this article can be found online at http://dx.doi.org/10.7717/peerj.3228#supplemental-information.

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
