# Peer review of "A new molecular diagnostic tool for surveying and monitoring Triops cancriformis populations"

_PeerJ, doi:10.7717/peerj.3228_

## Round 0.1 · original submission · Minor Revisions

All three reviewers recommended Minor Revisions. Please revise your manuscript according to their suggestions.

Reviewer 1 ·

Basic reporting

The authors are clear and report copious data.

Experimental design

The experiment was well designed and rigorously conducted. The methods were meticulous.

Validity of the findings

I was thoroughly convinced that the authors found a viable way to assess T. cancriformis egg banks in the wild.

Additional comments

See attached.

Annotated reviews are not available for download in order to protect the identity of reviewers who chose to remain anonymous.

·

Basic reporting

The manuscript compares three main methods to monitor a charismatic and endangered species of freshwater crustaceans, typical of ephemeral water pools.
The study is clear, interesting, supported, and potentially useful also as a reference for other species.

Experimental design

Good and reliable experimental design and analytical settings.

Validity of the findings

The manuscript provides compelling evidence in the comparison between the methods, and provides a well-thought inference on the time and costs associated to each method.
In order to provide more convincing support for some sentences, I suggest performing additional statistical tests. In detail:
- Line 356-362: it would be good to test if the differences were significant or not, especially to support the statement that potentially extinct sites had a higher proportion of deteriorated eggs; the same holds true for the statements on line 470-492.
- Line 377-378: it would be good to test for the significance of the differences to be able to state “depending on the method”.
- Line 524-527: it would be good to provide a list of the other taxa that have been identified through the BLAST comparison of COI, so that all readers can be convinced by the reliability of the explanations; especially because ‘microorganisms’ may simply mean bacteria for most readers.

Additional comments

I can highlight only few minor issues to further improve the strength of the manuscript.

Line 33: provide author and year of the species at its first citation.
Line 62-65: the sentence is awkward to read.
Line 78: ‘for’ repeated twice.
Line 189: 7 days of hydroperiod, whereas figure 1 and lines 202-203, and 220 report 8 days.
Line 202: provide a reference for HotShot DNA extraction at its first citation.
Line 223: I am not sure that ‘destroyed’ is the correct term for the preservation in ethanol.
Line 294: five long sequences mentioned in the methods, but only four are mentioned in the results on line 394.
Line 614 and figure 1: T. Cancriformis -> T. cancriformis.

·

Basic reporting

The article provides novel monitoring method. Well and clearly written and referenced. Good job!

Experimental design

Design-related comments that sould be addressed in the MM section:
1) How uniformly were the samples taken from within the boundaries of the ponds? Were the samples from middle and marginal parts of individual pools considered equal? Of course, I am mentioning this, because highest concentrations of Triops, and possibly their eggs(?), are regularly present in central parts of the pools shortly before they dry out. Was this phenomenon considered in sampling design?
2) There is no clear indication, whether the pools were dry, or (to same degree) full at the time of sampling. In the latter case, the places holding highest eggs concentrations could theoretically remain unsampled.
3) Are there some comparative environmental data available for individual pools? Especially character and density of vegetation, physical disturbances (erosion, movement of animals or vehicles etc.), and intensity of grazing might be important factors influencing quality of habitat and possibly Triops stocks in individual pools. (According to my experience, undisturbed pools, which are subject to succession, i.e. overgrowing by plants, accumulation of plant debris and other sediments etc., tend to lose their Triops populations very fast, no matter how numerous the populations are.) I can imagine the environmental factors might be subject of another partial study, but am missing at least basic info on this in the presented manuscript.

Validity of the findings

Sound

Additional comments

1) I generally consider presentation of GPS locations in supplementary materials only rather unsuitable. Provision of coordinates in "Sample collection and preparation section" would be more convenient...or alternatively in Figure 4 after its slight modification – for example under individual letters in the right column (after widening the column a bit). In both cases not much or no extra space would be taken.
2) I noticed double „for“ in L78

---

## Round 0.2 · accepted · Accept

Thank you for imrproving your manuscript.